# The prevalence of mental disorders among homeless people in high-income countries: An updated systematic review and meta-regression analysis

**Stefan Gutwinski**[1☯], **Stefanie Schreiter**[1,2☯], **Karl Deutscher**[1], **Seena Fazel**[3]*

**1** Department of Psychiatry and Psychotherapy, Charité–Universitätsmedizin Berlin, Freie Universität Berlin and Humboldt-Universität zu Berlin, Berlin, Germany, **2** Biomedical Innovation Academy, Berlin Institute of Health, Berlin, Germany, **3** Department of Psychiatry, University of Oxford, Oxford, United Kingdom

☯ These authors contributed equally to this work.
* Seena.Fazel@psych.ox.ac.uk

## Abstract

**Data Availability Statement:** All relevant data are within the manuscript and its Supporting Information files.

### Background

Homelessness continues to be a pressing public health concern in many countries, and mental disorders in homeless persons contribute to their high rates of morbidity and mortality. Many primary studies have estimated prevalence rates for mental disorders in homeless individuals. We conducted a systematic review and meta-analysis of studies on the prevalence of any mental disorder and major psychiatric diagnoses in clearly defined homeless populations in any high-income country.

### Methods and findings

We systematically searched for observational studies that estimated prevalence rates of mental disorders in samples of homeless individuals, using Medline, Embase, PsycInfo, and Google Scholar. We updated a previous systematic review and meta-analysis conducted in 2007, and searched until 1 April 2021. Studies were included if they sampled exclusively homeless persons, diagnosed mental disorders by standardized criteria using validated methods, provided point or up to 12-month prevalence rates, and were conducted in high-income countries. We identified 39 publications with a total of 8,049 participants. Study quality was assessed using the JBI critical appraisal tool for prevalence studies and a risk of bias tool. Random effects meta-analyses of prevalence rates were conducted, and heterogeneity was assessed by meta-regression analyses. The mean prevalence of any current mental disorder was estimated at 76.2% (95% CI 64.0% to 86.6%). The most common diagnostic categories were alcohol use disorders, at 36.7% (95% CI 27.7% to 46.2%), and drug use disorders, at 21.7% (95% CI 13.1% to 31.7%), followed by schizophrenia spectrum disorders (12.4% [95% CI 9.5% to 15.7%]) and major depression (12.6% [95% CI 8.0% to 18.2%]). We found substantial heterogeneity in prevalence rates between studies, which was partially explained by sampling method, study location, and the sex distribution of

**Funding:** The Wellcome Trust (https://wellcome.org) granted the submission fee for this review to SF (grant number 202836/Z/16/Z). The funders had no role in study design, data collection and analysis, decision to publish, or preparation of the manuscript.

**Competing interests:** The authors have declared that no competing interests exist.

**Abbreviations:** CI, confidence interval; DSM, Diagnostic and Statistical Manual of Mental Disorders; PI, prediction interval.

participants. Limitations included lack of information on certain subpopulations (e.g., women and immigrants) and unmet healthcare needs.

## Conclusions

Public health and policy interventions to improve the health of homeless persons should consider the pattern and extent of psychiatric morbidity. Our findings suggest that the burden of psychiatric morbidity in homeless persons is substantial, and should lead to regular reviews of how healthcare services assess, treat, and follow up homeless people. The high burden of substance use disorders and schizophrenia spectrum disorders need particular attention in service development. This systematic review and meta-analysis has been registered with PROSPERO (CRD42018085216).

## Trial registration

PROSPERO CRD42018085216.

---

### Author summary

#### Why was this study done?

- Homelessness continues to affect a large number of people in high-income countries and is associated with an increased risk of mental disorders.

- To guide service development, further research, and public policy, reliable estimates on the prevalence of mental disorders among homeless individuals are needed.

- Many primary investigations into rates of mental disorders have been published since a previous comprehensive quantitative synthesis in 2008.

#### What did the researchers do and find?

- We performed a systematic database search, extracted data from primary reports, and assessed their risk of bias, resulting in a sample of 39 studies including information from over 8,000 homeless individuals in 11 countries.

- We conducted random effects meta-analyses of 7 common diagnostic categories. Prevalence estimates were all increased in homeless individuals compared with those in the general population. Alcohol use disorders had the highest absolute rate, at 37%, with substantially elevated proportional excesses compared to the general population for schizophrenia spectrum disorders and drug use disorders as well.

- There was substantial between-study variation in prevalence estimates, and meta-regression analyses found that sampling method, participant sex distribution, and study country explained some of the heterogeneity.

### What do these findings mean?

- The high burden of substance use disorders and severe mental illness in homeless people represents a unique challenge to public health and policy.

- Future research should prioritize quantification of unmet healthcare needs, and how they can be identified and effectively treated. Research on subgroups, including younger people and immigrant populations, is a priority for prevalence work.

## Introduction

Homelessness is recognized by the United Nations Economic and Social Council as an issue of global importance [1]. In high-income countries, around 2 million people have been homeless over the past decade [2]. In the US, the lifetime prevalence of homelessness is estimated at 4.2% [3], with around 550,000 individuals lacking fixed, regular, and adequate residence on any given night [4]. Patterns over time have differed by country, although homelessness has increased in many high-income countries in recent years, including in the US and UK since 2017 [2].

There has been an increasing recognition of the public health importance of homeless persons, with many studies reporting high rates of acute hospitalization, chronic diseases, and mortality [5–13]. Comorbidities increase these risks, particularly mental disorder comorbidities. For example, in a Danish population study, comorbidity of psychiatric disorders increased mortality rates by 70% [14]. Furthermore, mental illness among homeless individuals has been associated with elevated rates of criminal behavior and victimization [15,16], prolonged courses of homelessness [17,18], and perceived discrimination [19]. Mental disorders among homeless individuals are mostly treatable and represent an important opportunity to address health inequalities.

Information on the overall extent and pattern of mental disorders among homeless people is necessary to inform resource allocation and service development, and to allow researchers, clinicians, and policymakers to consider evidence gaps. The large number of primary studies, of varying quality and samples, means that systematic reviews are required to clarify and synthesize the evidence, underscore main findings, and consider implications. According to a recent umbrella review, there have been at least 7 systematic reviews with quantitative data synthesis in the past 2 decades [20]; however, most of them focused on individual diagnostic categories [21–24], examined specific age bands [24,25], or were limited to a single country [26]. The last meta-analysis to our knowledge that provided a comprehensive account of the prevalence of major mental disorders in homeless adults in high-income countries completed its search in 2007 [27], and since then, a considerable number of primary studies have been published [28,29]. Thus, we conducted an updated systematic review and meta-analysis on the prevalence of mental disorders among homeless people in high-income countries, and added the diagnostic categories of any mental disorder and bipolar disorder.

## Methods

### Search strategy

We searched for studies that determined prevalence rates for at least 1 of the following disorders among homeless persons: (1) schizophrenia spectrum disorders, (2) major depressive

disorder, (3) bipolar disorder, (4) alcohol use disorders, (5) drug use disorders, (6) personality disorders, and (7) any current mental disorder (Axis I disorders in the Diagnostic and Statistical Manual of Mental Disorders [DSM] multiaxial system [30]).

We have updated an earlier review [27] that was based on a search for articles published up until December 2007, so we targeted new primary studies published between 1 January 2008 and 1 April 2021. We searched Embase via OvidSP, Medline via OvidSP and via PubMed, and PsycInfo via EBSCOhost. Additionally, we searched Google Scholar using a search query and screened all literature citing the previous review. Finally, we screened reference lists of relevant publications. Each search employed a specific combination of search terms designed to fit the databases' respective syntaxes and thesaurus systems (S1 Table). Articles written in languages other than English or German were translated by professional translators. The protocol for this systematic review and meta-analysis has been published (PROSPERO CRD42018085216). We followed Meta-analysis of Observational Studies in Epidemiology (MOOSE) guidelines for extracting and assessing data [31]. This systematic review adheres to the Preferred Reporting Items for Systematic Reviews and Meta-Analyses (PRISMA) statement (see S2 Table) [32].

## Eligibility criteria and study selection

Inclusion criteria were as follows: (1) homelessness status of study participants was validated by an operationalized definition or a sampling method that specifically targeted homeless population; (2) standardized criteria for the psychiatric disorders specified above, based on the International Classification of Diseases (ICD) or DSM, were applied; (3) psychiatric diagnoses were made by clinical examination or interviews using validated semi-structured diagnostic instruments; (4) for any psychiatric disorders except for personality disorders (where lifetime rates were used), prevalence rates were reported within 12 months; and (5) study location was a high-income country according to the classification of the World Bank [33].

Surveys that reported a response rate of less than 50% or exclusively sampled from selected subpopulations (such as elderly homeless, homeless youth, or homeless single parents) were excluded.

In order to assess all results from the bibliographic search process, researchers SS, SG, and KD each carried out a multilevel screening process independently from one another. Any differences between results were resolved by consensus between all the authors.

## Data collection and quality assessment

Information from included surveys was extracted on study location, year of diagnostic assessment, operational definition of homelessness status, sampling method, diagnostic procedures, diagnostic criteria, professional qualification of interviewers, response rate, dropout rate, number of participants by sex, sample mean age, current accommodation of participants, sample mean duration of homelessness, and number of participants diagnosed with schizophrenia spectrum disorders, major depressive disorder, bipolar disorder, alcohol- and drug-related disorders, personality disorders, and any primary diagnosis of a mental disorder apart from personality and developmental disorders (i.e., Axis I disorders in DSM). If data regarding any of these categories were unclear in the published study, we corresponded with the primary study authors.

Each included publication was rated on methodological quality by 2 sets of criteria specifically designed to assess prevalence studies: the JBI critical appraisal tool for prevalence studies [34] and a risk of bias tool [35]. This process was carried out by SS, SG, and KD independently, and any differences were resolved by discussion.

## Statistical analysis

Random effects meta-analyses and meta-regression analyses were performed on each diagnostic category independently—prevalence data for alcohol misuse/abuse and alcohol dependence were both entered into the single category of alcohol use disorders, in accordance with current diagnostic approaches. All analyses were done in R, version 4.0.4 [36]. The package "metafor," version 2.4–0, was utilized for meta-analysis and meta-regression analysis, supplemented by "glmulti," version 1.0.8, for multivariable model selection and "mice", version 3.13.0, for multivariate imputation [37–39].

Prevalence estimates were transformed on the double arcsine function in order to avoid variance instability and confidence intervals (CIs) exceeding the interval ($0 \leq x \leq 1$) in which prevalence proportions can be meaningfully defined [40]. We calculated random effects models, which we deemed appropriate considering sampling differences. The Paule–Mandel estimator was chosen to measure between-study variance due to its reliability for different types of models [41]. A $Q$-test for heterogeneity was conducted. To quantify measures of between-study heterogeneity, we report the test statistic $Q_E$ and corresponding $p$-value as well as the $I^2$ statistic. Additionally, we calculated 95% prediction intervals (PIs) for all meta-analytical models [42]. Because the "metafor::predict.rma" function unrealistically assumed that the model variance $\tau^2$ was a known value [43], we instead implemented a method proposed by Higgins and colleagues that accounts for $\tau^2$ being an estimate with limited precision ([44], expression 12).

Additional meta-analyses were carried out in each diagnostic category for low-risk-of-bias studies, assigned during quality assessment [35]. Subgroup analyses comparing low-risk-of-bias and moderate-risk-of-bias studies were performed through a $Q$-test. In cases of significant between-subgroup difference, a meta-regression model with risk of bias assessment as a single independent variable was computed to estimate the proportion of variance explained by disparities in methodological quality.

For each diagnostic category, meta-regression analyses were performed to investigate potential sources of heterogeneity. Continuous independent variables for single factor meta-regression were number of participants, sex distribution (female/all), and final year of diagnostic assessments. Categorial independent factors were diagnostic method (structured/semi-structured interview versus non-structured clinical evaluation), sampling method (randomized versus non-randomized sampling methods), and study location (US, UK, or Germany). The 3 study locations were prespecified as predictor variables due to a preponderance of primary studies in each of these countries.

Multivariable meta-regression models were also calculated. The respective independent variables were chosen through automated, information-criterion-based model selection with generalized linear models [38]. For models with 20 or more included studies, the Akaike information criterion (AIC) was used; for models with fewer than 20 included studies, we utilized the corrected version for small sample sizes ($AIC_C$) to avoid over-fitting.

The proportion of variance of prevalence estimates explained by any meta-regression model was estimated by the $R^2$ statistic [45].

We assumed that missingness was at random [46], so missing values in independent variables (that were missing despite requests for additional information to primary study authors) were replaced through multiple imputation by chained equations [47]. For models including incomplete predictor variables, results of meta-regression on imputed data are presented as the primary analysis; meta-regression results on only complete cases are provided as sensitivity analyses [48].

## Results

### Description of included studies

The systematic literature search returned 5,886 distinct records, of which 144 full texts were assessed (see S3 Table for reasons for exclusion). We identified a total of 39 studies comprising data on 8,049 homeless individuals [28,29,49–85] (see Fig 1 for flow chart of screening process). This included 10 additional studies for this update [28,29,53–55,57,59,62,75,76], and 2 previous investigations were further clarified [81,83].

Out of the 39 included studies, 27 publications reported age (mean of 41.1 years) [28,49–57,59,64,65,67,70–73,75,76,78–82,84,85], and the proportion of women was 22.3% (based on 38 studies) [28,29,49–65,67–85]. Eleven studies [52,56,68,71–74,80,82,83,85] investigated male-only samples, and 5 studies [59,65,78,81,84] solely women. Of the 39 studies, 27 were from 3 countries: 11 from the US (n = 2,694 participants) [57,59,60,64,69,75,77,79,83–85], 7 from the UK (n = 1,390 participants) [49,61,66,67,70,78,82], and 9 from Germany (n = 936 participants) [28,50,51,56,65,71–73,80,81]. Six studies were from other European countries (n = 2,301 participants) [29,52,55,58,62,76], 1 study was from Canada (n = 60 participants) [69], 2 were from Japan (n = 194 participants) [53,54], and 2 were from Australia (n = 667 participants) [63,74]. Fourteen studies reported a response rate of 85% or above [49,60,65,66,69–73,78,81,82,84,85], 20 studies reported a response rate below 85% [28,29,50–52,54–56,58,59,61–64,68,74,76,79,80,83], and 5 did not report participation rate [53,57,67,75,77]. In 13 studies, participants were accommodated in shelters, hostels, or residential care when assessed [28,49,63,66,70,72,74,76–78,81–83] while in 3 they were rough sleeping [54,62,67]; 22 studies had mixed samples regarding accommodation or provided incomplete information [29,50–53,55–61,64,65,68,69,71,73,75,79,80,84,85]. S4 Table provides further information on methodological and sample characteristics. For quality ratings, see S5 and S6 Tables. S7 Table provides all extracted data that meta-analyses and meta-regression analyses were based on.

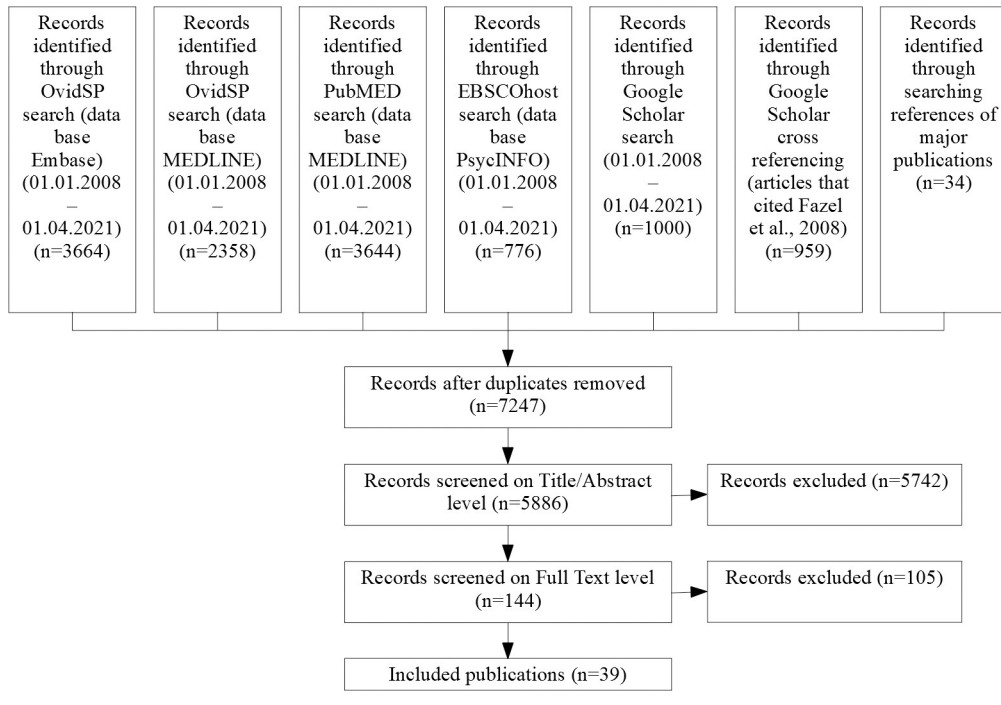

**Fig 1. Screening process.**

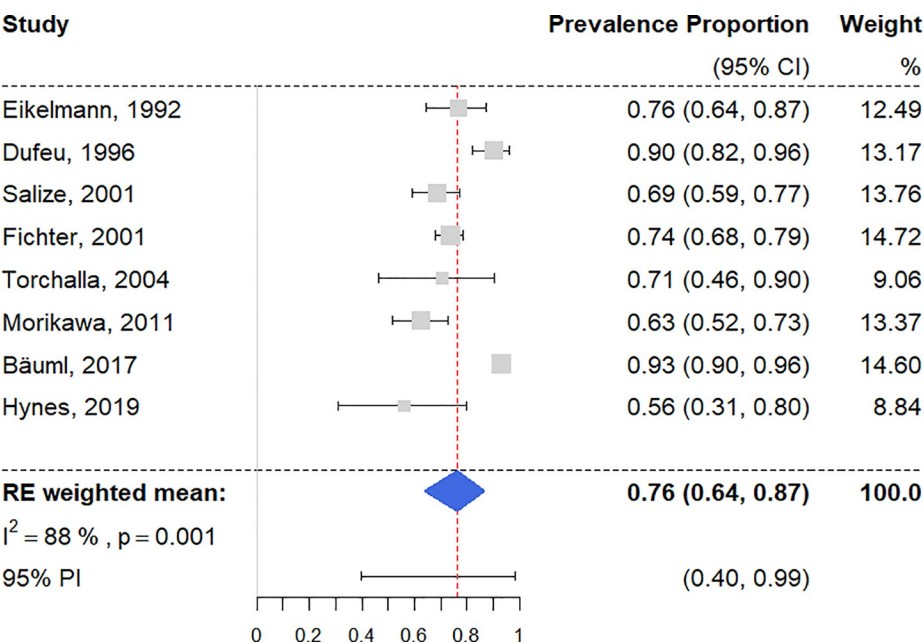

**Fig 2. Forest plot of prevalence estimates of any current mental disorder.** Analytic weights are from random effects meta-analysis. Grey boxes represent study estimates; their size is proportional to the respective analytical weight. Lines through the boxes represent the 95% CIs around the study estimates. The blue diamond represents the mean estimate and its 95% CI. The vertical red dashed line indicates the mean estimate. CI, confidence interval; PI, prediction interval.

## Any current mental disorders

There were 8 surveys reporting on homeless people having at least 1 diagnosis of a current mental disorder [28,51,54,62,71–73,81], with a random effects pooled prevalence estimated at 76.2% (95% CI 64.0% to 86.6%) (Fig 2). Individual study prevalence rates ranged from 56.3% to 93.3%, with substantial heterogeneity ($I^2$ = 88% [95% CI 72% to 97%]). The 95% PI was 40% to 99%. Univariable meta-regression analysis revealed that studies with randomized sampling procedures reported significantly higher prevalence estimates than ones with other sampling procedures, accounting for a large proportion of heterogeneity ($R^2$ = 59%) (see S8 Table). Sampling procedure was chosen as the only predictor variable by multivariable model selection (see Table 1).

In a subgroup analysis of 4 low-risk-of-bias studies [62,71,73,81], the random effects prevalence was 75.3% (95% CI 50.2% to 93.6%; $I^2$ = 81% [95% CI 32% to 99%]). There was no significant difference between quality subgroups ($Q$ = 0.03, $p$ = 0.87).

## Schizophrenia spectrum disorders

There were 35 surveys reporting on any schizophrenia spectrum disorder [28,29,49,51–58,60–74,76–78,80–85], and the random effects prevalence was 12.4% (95% CI 9.5% to 15.7%) (Fig 3), with substantial heterogeneity ($I^2$ = 93% [95% CI 89% to 96%]; 95% PI 0% to 34%). Primary investigation estimates ranged between 2.0% and 42.2%. No single model coefficient in univariable meta-regression was statistically significant. A multivariable model with sample size, proportion of female participants, and study location in Germany accounted for a small share of the heterogeneity ($R^2$ = 16%). The latter model indicated that studies with smaller samples had significantly higher prevalence rates, but only when based on imputed values (see Table 1).

**Table 1. Study-level factors associated with between-study heterogeneity in multivariable meta-regression.**

| Model | β (standard error) and p-value, by factor | | | | | | |
|---|---|---|---|---|---|---|---|
| | Any current mental disorder[a,b] | Schizophrenia spectrum disorders | Major depression | Bipolar disorder | Alcohol use disorders[b] | Drug use disorders | Personality disorders[a,b] |
| **Imputed model** | — | *With $R^2$ = 16%* | *With $R^2$ = 32%* | *With $R^2$ = 54%* | — | *With $R^2$ = 42%* | — |
| | — | Sample size (continuous): >**−0.01 (<0.01)** ***p* = 0.04** | Randomized sampling versus other: **0.17 (0.07)** ***p* = 0.03** | Randomized sampling versus other: **−0.11 (0.04)** ***p* = 0.03** | — | Randomized sampling versus other: **−0.39 (0.12)** ***p* < 0.01** | — |
| | | Sex distribution (female/all): 0.13 (0.07) *p* = 0.09 | Sex distribution (female/all): 0.14 (0.09) *p* = 0.15 | Sex distribution (female/all): **0.15 (0.06)** ***p* = 0.04** | | UK versus other locations: **−0.48 (0.18)** ***p* = 0.01** | |
| | | Germany versus other locations: −0.09 (0.05) *p* = 0.11 | | | | | |
| **Complete case analysis** | *With $R^2$ = 59%* | *With $R^2$ = 13%* | *With $R^2$ = 32%* | *With $R^2$ = 50%* | *With $R^2$ = 27%* | *With $R^2$ = 51%* | *With $R^2$ = 15%* |
| | Randomized sampling versus other: **0.23 (0.08)** ***p* = 0.03** | Sample size (continuous): >−0.01 (<0.01) *p* = 0.07 | Randomized sampling versus other: **0.18 (0.08)** ***p* = 0.03** | Randomized sampling versus other: **−0.12 (0.04)** ***p* = 0.03** | Germany versus other locations: **0.33 (0.10)** ***p* < 0.01** | Randomized sampling versus other: **−0.39 (0.10)** ***p* < 0.01** | North America versus other locations: 0.30 (0.17) *p* = 0.10 |
| | | Sex distribution (female/all): 0.14 (0.08) *p* = 0.08 | Sex distribution (female/all): 0.18 (0.10) *p* = 0.10 | Sex distribution (female/all): 0.12 (0.06) *p* = 0.09 | North America versus other locations: **0.20 (0.10)** ***p* = 0.04** | UK versus other locations: **−0.72 (0.20)** ***p* < 0.01** | |
| | | Germany versus other locations: −0.09 (0.05) *p* = 0.11 | | | | | |

Statistically significant values given in bold.

[a]A univariable model was chosen in model selection, so only 1 variable is presented.

[b]All variables in the multivariable model were complete, so no imputation was needed.

A subgroup analysis of 17 low-risk-of-bias studies [29,49,52,55,58,60,62,65,67,69,71,73,78, 80,81,84,85] revealed a random effects pooled prevalence of 10.5% (95% CI 6.2% to 15.7%; $I^2$ = 94% [95% CI 88% to 98%]). The subgroup difference between low-risk-of-bias and moderate-risk-of-bias studies was non-significant, with the low-risk group resulting in a marginally lower weighted mean ($Q$ = 1.59, *p* = 0.21).

## Major depression

We identified 18 studies reporting prevalence estimates on major depressive disorder [28,49,52,55,57–60,62,63,65,67,71,77,80,81,84,85], with a random effects pooled prevalence of 12.6% (95% CI 7.9% to 18.2%) (Fig 4). Individual study estimates ranged between 0% and 40.6% and showed substantial heterogeneity ($I^2$ = 95% [95% CI 90% to 98%]; 95% PI 0% to 40%). Univariable meta-regression analysis produced no significant models (see S8 Table). For multivariable regression, independent variable sampling procedure and proportion of female participants were selected; the model indicated that studies with randomized sampling reported significantly higher prevalence rates (see Table 1).

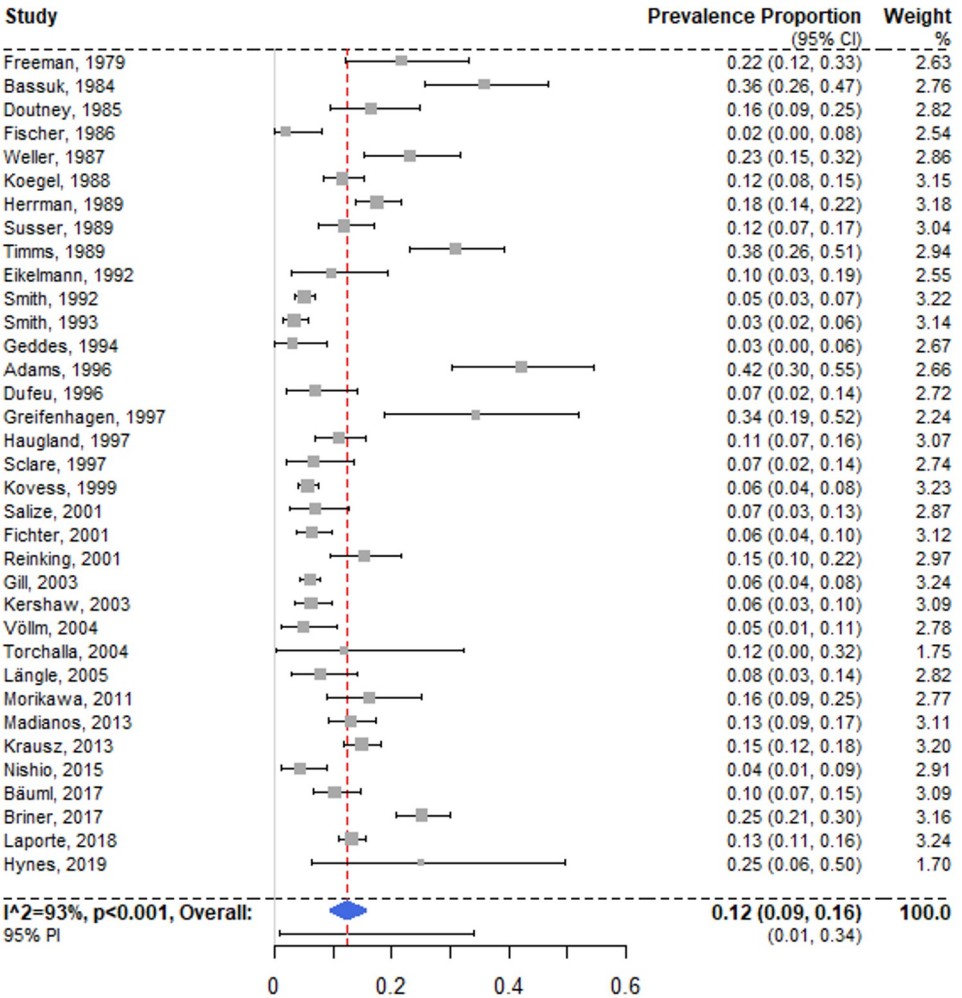

**Fig 3. Forest plot of prevalence estimates of schizophrenia spectrum disorders.** Analytic weights are from random effects meta-analysis. Grey boxes represent study estimates; their size is proportional to the respective analytical weight. Lines through the boxes represent the 95% CIs around the study estimates. The blue diamond represents the mean estimate and its 95% CI. The vertical red dashed line indicates the mean estimate. CI, confidence interval; PI, prediction interval; RE, random effects.

In a subgroup analysis of 13 low-risk-of-bias studies [49,52,55,58,60,62,65,67,71,80,81, 84,85], the random effects pooled prevalence was 13.0% (95% CI 6.7% to 20.9%; $I^2$ = 96% [95% CI 90% to 99%]). There were no significant differences in between risk of bias subgroups ($Q$ = 0.09, $p$ = 0.76).

## Bipolar disorder

Fourteen surveys with prevalence estimates on bipolar disorder were identified [28,49,55,57–59,62,63,65,67,71,77,84,85]. Three studies reported on solely type I bipolar disorder [49,57,85], 4 examined all bipolar disorder subtypes [28,59,65,71], and 7 did not specify [55,58,62,63,67,77,84]. The random effects pooled prevalence was 4.1% (95% CI 2.0% to 6.7%) (Fig 5), with substantial heterogeneity ($I^2$ = 89% [95% CI 77% to 96%]; 95% PI 0% to 16%). Individual estimates ranged from 1.0% to 13.5%. Univariable regression models indicated that studies with higher proportions of female participants reported significantly higher rates of

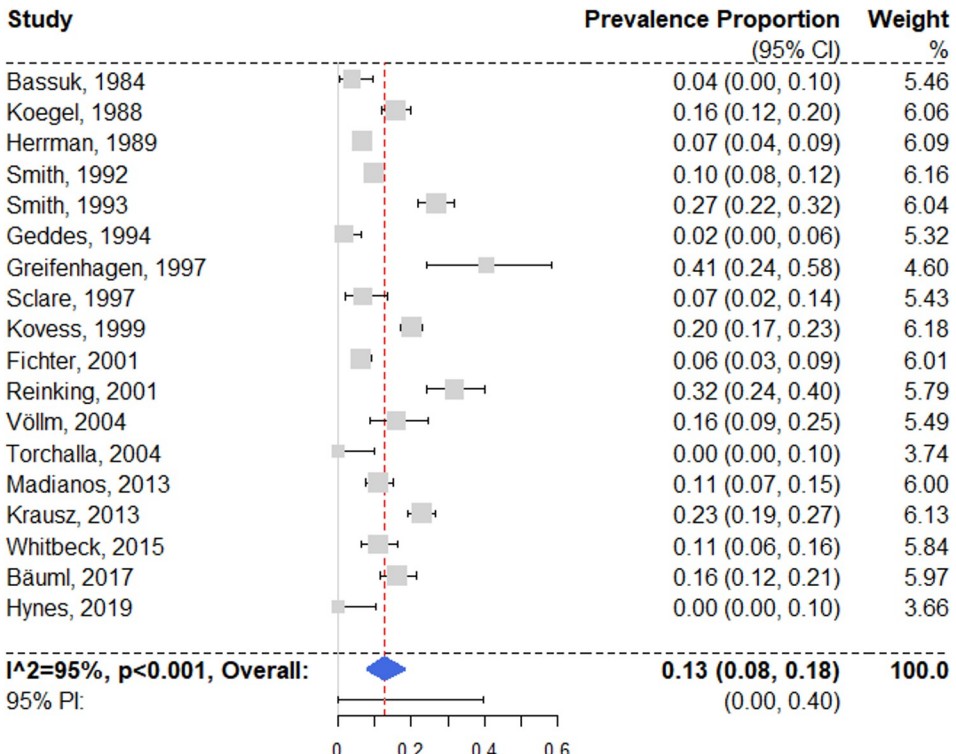

**Fig 4. Forest plot of prevalence estimates of major depression.** Analytic weights are from random effects meta-analysis. Grey boxes represent study estimates; their size is proportional to the respective analytical weight. Lines through the boxes represent the 95% CIs around the study estimates. The blue diamond represents the mean estimate and its 95% CI. The vertical red dashed line indicates the mean estimate. CI, confidence interval; PI, prediction interval; RE, random effects.

bipolar disorder (see S8 Table). In the multivariable model, prevalence estimates from studies with randomized sampling were significantly lower than those from studies with other sampling methods (see Table 1).

A subgroup analysis of 9 low-risk-of-bias surveys [49,55,58,62,65,67,71,84,85] resulted in a random effects pooled prevalence of 2.6% (95% CI 1.0% to 4.9%), with moderate heterogeneity ($I^2$ = 78% [95% CI 29% to 96%]). The difference between low-risk-of-bias and moderate-risk-of-bias studies was non-significant ($Q$ = 2.29, $p$ = 0.13).

Findings for any affective disorder (which included depression and bipolar disorder) are reported in S1 Text and S6 Table.

## Alcohol use disorders

Estimates on alcohol use disorders could be extracted from 29 surveys [28,29,51–66,68,71–73,76,77,79–81,84,85]. The random effects pooled prevalence was 36.7% (95% CI 27.7% to 46.2%) (Fig 6), with individual study estimates ranging from 5.5% to 71.7%, and with substantial between-study heterogeneity ($I^2$ = 98% [95% CI 97% to 99%]; 95% PI 2% to 85%). Univariable meta-regression models indicated that studies with smaller samples and studies from Germany (compared to other locations) reported significantly higher rates of alcohol use disorders (see S8 Table). In multivariable analysis, the best selected model included only study location as a predictor variable, with higher prevalences reported in Germany and North America (see Table 1).

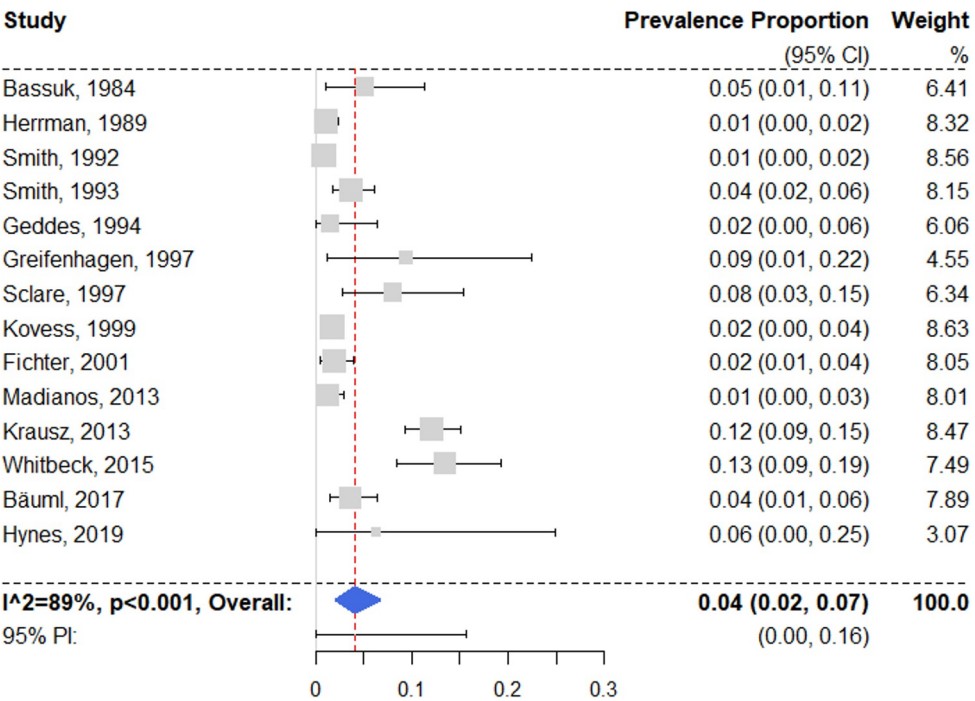

**Fig 5. Forest plot of prevalence estimates of bipolar disorder.** Analytic weights are from random effects meta-analysis. Grey boxes represent study estimates; their size is proportional to the respective analytical weight. Lines through the boxes represent the 95% CIs around the study estimates. The blue diamond represents the mean estimate and its 95% CI. The vertical red dashed line indicates the mean estimate. CI, confidence interval; PI, prediction interval; RE, random effects.

In a subgroup analysis of 14 low-risk-of-bias studies [29,52,55,58,60,62,65,71,73,79–81,84,85], the random effects pooled prevalence was 36.9% (95% CI 21.1% to 54.3%; $I^2$ = 99% [95% CI 98% to 100%]). There was no significant difference between risk of bias subgroups ($Q < 0.01$, $p = 0.96$).

## Drug use disorders

We identified 23 surveys reporting prevalence estimates on drug use disorders [28,29,52,53,55–65,71,73,76,79,80,82,84,85] (Fig 7). A random effects pooled prevalence of 21.7% (95% CI 13.1% to 31.7%) was found, with very high heterogeneity ($I^2$ = 99% [95% CI 98% to 99%]; 95% PI 0% to 74%); individual estimates ranged between 0% and 72.1%. According to univariable meta-regression, studies with randomized sampling (as opposed to other sampling methods) estimated significantly lower prevalence rates (see S8 Table). The selected multivariable model showed that studies from the UK reported lower prevalence rates. These results were confirmed by a secondary complete case analysis.

A subgroup analysis of 13 low-risk-of-bias studies [29,52,55,58,60,62,65,71,73,79,80,84,85] resulted in a random effects pooled prevalence of 18.1% (95% CI 10.5% to 27.2%), with substantial heterogeneity ($I^2$ = 97% [95% CI 94% to 99%]). The difference between subgroups was not significant ($Q = 0.65$, $p = 0.42$).

## Personality disorders

Fourteen studies reported prevalence estimates on lifetime personality disorders [28,51–53,62,64,67,75–77,80,82,84,85], with a random effects pooled prevalence of 25.4% (95% CI

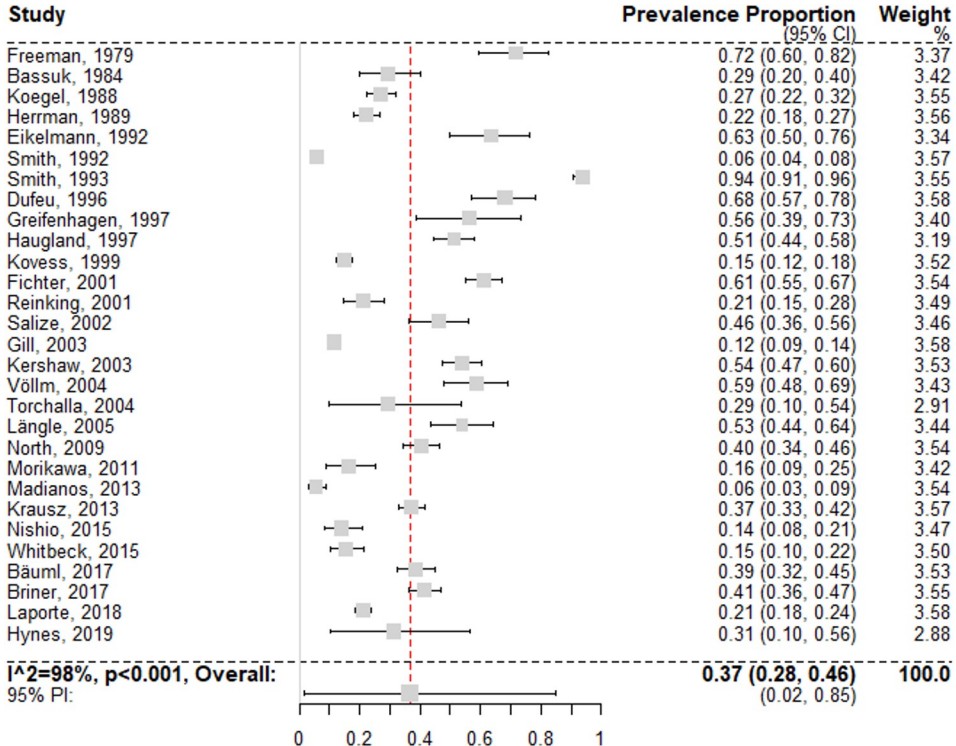

**Fig 6. Forest plot of prevalence estimates of alcohol use disorders.** Analytic weights are from random effects meta-analysis. Grey boxes represent study estimates; their size is proportional to the respective analytical weight. Lines through the boxes represent the 95% CIs around the study estimates. The blue diamond represents the mean estimate and its 95% CI. The vertical red dashed line indicates the mean estimate. CI, confidence interval; PI, prediction interval; RE, random effects.

10.9% to 43.6%) (Fig 8). Individual estimates ranged between 0% and 98.3%, resulting in substantial heterogeneity ($I^2$ = 99% [95% CI 97% to 99%]; 95% PI 0% to 91%). Univariable regression models did not yield significant results (see S8 Table), and neither did the selected multivariable model (see Table 1).

In a subgroup analysis of 6 low-risk-of-bias studies [52,62,67,80,84,85], the random effects pooled prevalence was 21.0% (95% CI 4.7% to 44.5%), with substantial heterogeneity ($I^2$ = 97% [95% CI 92% to 100%]). The difference between subgroups was not significant ($Q$ = 0.32, $p$ = 0.57).

## Discussion

This systematic review and meta-analysis of the prevalence of mental illness among homeless people in high-income countries included 39 studies comprising a total of 8,049 participants. We investigated 7 common psychiatric diagnoses, and examined possible explanations for the between-study heterogeneity. We report 3 main findings.

With a pooled prevalence of around 37%, alcohol-related disorders were the most prevalent diagnostic category. This prevalence estimate is around 10-fold greater than general population estimates: An EU study reported a 12-month prevalence of 3.4% in the general population [86]. Correspondingly, drug-related disorders were the second most common current mental disorder, with a pooled prevalence of 22% (which can be compared with the 12-month prevalence in the US general population of 2.5% [87]). We found substantial variation between the

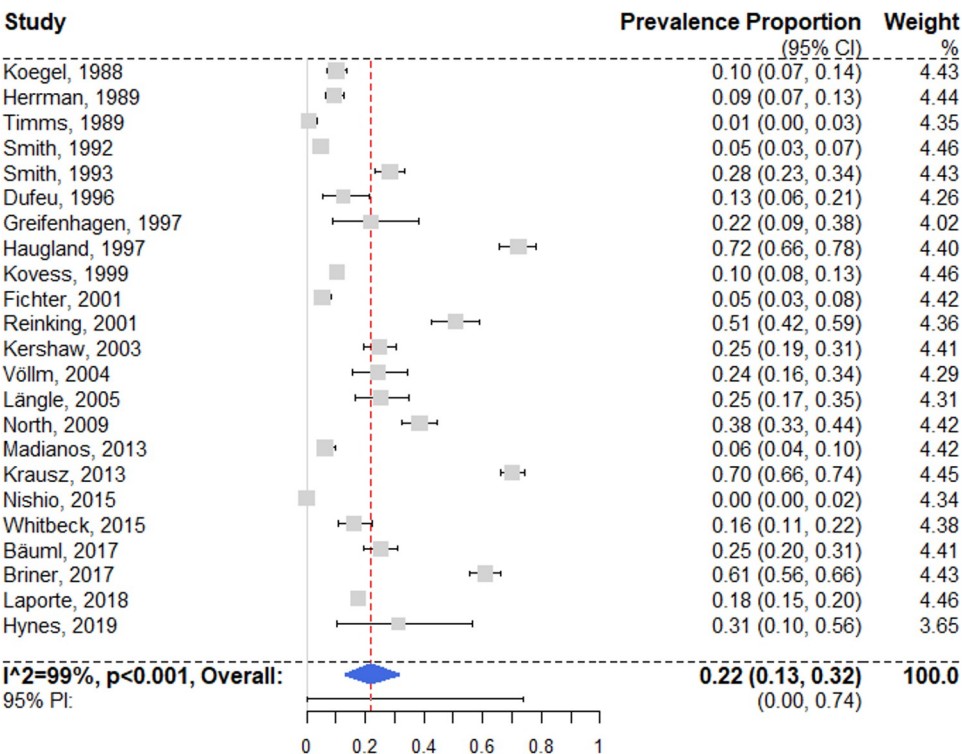

**Fig 7. Forest plot of prevalence estimates of drug use disorders.** Analytic weights are from random effects meta-analysis. Grey boxes represent study estimates; their size is proportional to the respective analytical weight. Lines through the boxes represent the 95% CIs around the study estimates. The blue diamond represents the mean estimate and its 95% CI. The vertical red dashed line indicates the mean estimate. CI, confidence interval; PI, prediction interval; RE: random effects.

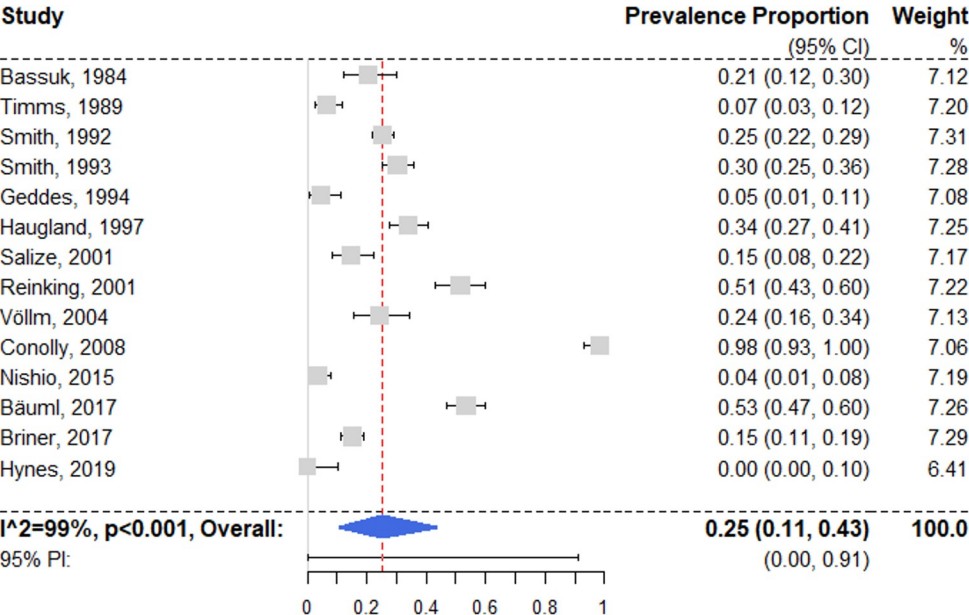

**Fig 8. Forest plot of prevalence of personality disorders.** Analytic weights are from random effects meta-analysis. Grey boxes represent study estimates; their size is proportional to the respective analytical weight. Lines through the boxes represent the 95% CIs around the study estimates. The blue diamond represents the mean estimate and its 95% CI. The vertical red dashed line indicates the mean estimate. CI, confidence interval; PI, prediction interval; RE, random effects.

individual studies contributing to these estimates, with individual study estimates ranging from 5.5% to 71.7% for alcohol-related disorders; this variation was partially accounted for by study location. Particularly, German-based samples typically had higher prevalence rates of alcohol use disorders than those from other nations. This might highlight geographical differences regarding the affordability and availability of substances, including a comparatively low alcohol tax in Germany [88]. Irrespective of this moderating factor, the strong association between homelessness and substance abuse reflects a bidirectional relationship: Alcohol and drug use represent possible coping strategies in marginalized housing situations. At the same time, substance abuse and other psychiatric disorders precede the onset of homelessness in many people, with alcohol use disorders in particular emerging at an earlier point in life compared to age-matched non-homeless comparisons [89], suggesting that substance use might contribute to the deterioration of an individual's housing situation. Such deterioration is consistent with the links between substance use disorders and excess mortality in homeless people [11], homelessness chronicity, psychosocial problems [90], and poorer long-term housing stability [91].

A second main finding was that some study characteristics consistently explained the variations in prevalence. In 5 diagnostic groups, methods were important, specifically the number of included participants and the sampling procedure. Unexpectedly, the latter had differential effects by diagnostic group. In bipolar disorder and drug use disorders, randomization was associated with lower prevalence estimates, whereas for any current mental disorder and major depression, it was associated with higher estimates. These findings underline the importance of standardized methodological procedures for homelessness research. We recommend that new research studies should base their inclusion criteria on a standardized definition of homelessness based on ETHOS criteria [92] and use randomized sampling, standardized diagnostic instruments, and trained interviewers with clinical backgrounds (including nurses, psychologists, and medical doctors).

Our third main finding was high prevalence rates for treatable mental illnesses, with 1 in 8 homeless individuals having either major depression (12.6%) or schizophrenia spectrum disorders (12.4%). This represents a high rate of schizophrenia spectrum disorders among homeless people, and a very large excess compared to the 12-month prevalence in the general population, which for schizophrenia is estimated around 0.7% in high-income countries [86]. For major depression, the difference from the general population is not marked, as the 12-month prevalence in the US general population is estimated at 10% [93], although comparisons would need to account for the differences in age and sex structure between the samples contributing to this review and the general population. Depression remains important because it is modifiable, and because of its effects on adverse outcomes. In addition, a recent cohort study based in Vancouver, Canada, found that substance use disorders were associated with worsening of psychosis in homeless people, underscoring the links between these mental disorders, and the importance of treatment in mitigating their effects directly and indirectly [13]. This study also found elevated risks of mortality in those with psychosis and alcohol use disorders [13].

Overall, our findings underscore the importance of mental health problems among homeless individuals. This review is complemented by other research on the often precarious financial and housing situation of psychiatric patients, for whom high rates of homelessness, indebtedness, and lack of bank account ownership have been reported [94–97]. Being homeless and having mental disorders are therefore closely interrelated. Fragmented and siloed services will therefore be typically unable to address these linked psychosocial and health problems. The mental disorders reported in this study are typically associated with unmet needs in the homeless population [51,98–100], which further indicates the need for integrated

approaches. Many different initiatives to address these needs have been researched over the last decade, among them Housing First, Intensive Case Management, Assertive Community Treatment, and Critical Time Intervention. Randomized controlled studies using these approaches have generally resulted in positive effects on housing stability, but only moderate or no effects on most indicators of mental health in comparison to usual care, including for substance use [101–104]. Therefore, further improvements in management and treatment are necessary that focus on these common mental disorders.

The COVID-19 pandemic has put homeless people at particular risk of infection and further marginalization [105]. But it has shown what is possible—government agencies and charity organizations managed to quickly provide accommodation to a large number of rough-sleeping homeless people in some European regions [106,107].

Some limitations to this review need to be considered. We searched a limited number of databases, so it is possible that we missed certain primary reports, although this possibility was minimized by searching through reference lists and Google Scholar citations. Furthermore, despite the high rate of multimorbidity in homeless populations [108,109], included studies lacked information on comorbidity. With most of the primary studies reporting prevalence rates of more than 1 of the investigated diagnostic categories, effects from the same sample were in many cases entered into multiple meta-analytical models. This may have led to measurement error and overestimation if diagnostic criteria overlap, but without diagnostic validity studies specific to homeless persons, this remains uncertain. We limited the number of demographic variables that we conducted heterogeneity analyses on, because of variations in measurement and reporting detail. Future work, including individual participant meta-analysis, could standardize information on age, socioeconomic background, and ethnicity, for example.

The present review focuses on high-income countries because sample and diagnostic heterogeneity would presumably increase if a wider range of countries was included. It is important to note, however, that homeless populations in low- and middle-income countries need investigation, and may have higher rates of trauma-related symptoms [110,111]. The prevalence of the mental disorders reported in the current review does not consider unmet healthcare needs or treatment provision, which are additional elements to consider in developing services. Finally, several subpopulations were underrepresented: migrants and refugees (individuals who did not speak the local language were excluded from some study samples), the "hidden homeless" population (e.g., "couch-surfers") [112] (sampling procedures were often not able to identify this group), and, importantly, homeless women. Twenty-two percent of participants in the included studies were female, lower than most estimates of the proportion of women among homeless populations, which range between 25% and 40% [4,113].

In summary, we found high prevalence of mental disorders among homeless people in high-income countries, with around three-quarters having any mental disorder and a third having alcohol use disorders. Future research should focus on integrated service models addressing the identified needs of substance use disorders, schizophrenia spectrum disorders, and depression in homeless individuals as a priority. In addition, new work could consider focusing on underrepresented subpopulations like homeless women and migrants. Furthermore, longitudinal studies could examine mechanisms linking homelessness and mental disorders in order to develop more effective preventive measures.

## Supporting information

**S1 Table. Database search strings.**
(DOCX)

**S2 Table. PRISMA 2009 checklist.**
(DOCX)

**S3 Table. Studies excluded at full-text screening, with reasons.**
(DOCX)

**S4 Table. Study characteristics.**
(DOCX)

**S5 Table. JBI checklist for prevalence studies.**
(DOCX)

**S6 Table. Risk of bias tool.**
(DOCX)

**S7 Table. Data basis for meta-analyses and meta-regression analyses.**
(DOCX)

**S8 Table. Univariable regression models.**
(DOCX)

**S9 Table. Results of single factor meta-regression models for affective disorders (pooled).**
(DOCX)

**S10 Table. Results of multiple factor meta-regression for affective disorders (pooled).**
(DOCX)

**S1 Text. Affective disorders.** Results of meta-analysis and meta-regression analysis.
(DOCX)

## Acknowledgments

We are grateful to authors of included and non-included publications who provided additional details about their studies: C. Adams, H.-J. Salize, A. Greifenhagen, C. Vazquez, U. Beijer, C. Siegel, and G. Gilchrist. We are also grateful to the professional translator N. Spennemann for assistance with Japanese studies.

## Author Contributions

**Conceptualization:** Stefan Gutwinski, Stefanie Schreiter, Seena Fazel.

**Data curation:** Stefan Gutwinski, Karl Deutscher.

**Formal analysis:** Karl Deutscher.

**Funding acquisition:** Seena Fazel.

**Methodology:** Stefan Gutwinski, Stefanie Schreiter, Karl Deutscher, Seena Fazel.

**Project administration:** Stefan Gutwinski, Seena Fazel.

**Software:** Karl Deutscher.

**Supervision:** Stefan Gutwinski, Stefanie Schreiter, Seena Fazel.

**Visualization:** Karl Deutscher.

**Writing – original draft:** Stefan Gutwinski, Stefanie Schreiter, Karl Deutscher.

**Writing – review & editing:** Stefan Gutwinski, Stefanie Schreiter, Karl Deutscher, Seena Fazel.

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
