## [Editor Report · Decision Letter 0]

3 Feb 2021

Dear Dr Fazel, 

Thank you for submitting your manuscript entitled "The prevalence of mental disorders among homeless people in high-income

countries: updated systematic review and meta-regression analysis" for consideration by PLOS Medicine.

Your manuscript has now been evaluated by the PLOS Medicine editorial staff as well as by an academic editor with relevant expertise and I am writing to let you know that we would like to send your submission out for external peer review.

Kind regards,

Caitlin Moyer, Ph.D.

Associate Editor

PLOS Medicine

---

## [Decision Letter · Decision Letter 1]

18 Mar 2021

Dear Dr. Fazel,

Thank you very much for submitting your manuscript "The prevalence of mental disorders among homeless people in high-income countries: updated systematic review and meta-regression analysis" (PMEDICINE-D-21-00563R1) for consideration at PLOS Medicine. 

Your paper was evaluated by a senior editor and discussed among all the editors here. It was also sent to four independent reviewers, including a statistical reviewer. The reviews are appended at the bottom of this email and any accompanying reviewer attachments can be seen via the link below:

[LINK]

In light of these reviews, I am afraid that we will not be able to accept the manuscript for publication in the journal in its current form, but we would like to consider a revised version that addresses the reviewers' and editors' comments. Obviously we cannot make any decision about publication until we have seen the revised manuscript and your response, and we plan to seek re-review by one or more of the reviewers. 

We expect to receive your revised manuscript by Apr 08 2021 11:59PM. Please email us (plosmedicine@plos.org) if you have any questions or concerns.

We look forward to receiving your revised manuscript. 

Sincerely,

Caitlin Moyer, Ph.D.

Associate Editor 

PLOS Medicine

plosmedicine.org

1. Abstract: Please report your abstract according to PRISMA for abstracts, following the PLOS Medicine abstract structure (Background, Methods and Findings, Conclusions) http://www.plosmedicine.org/article/info:doi/10.1371/journal.pmed.1001419 .

2. Abstract Background: Provide the context of why the study is important. The final sentence should clearly state the study question.

3. Abstract: Methods and Findings: Please provide the dates of search, data sources, types of study designs included, eligibility criteria, and synthesis/appraisal methods.

4. Abstract: Methods and Findings: Please conclude this section with a sentence describing the main limitation(s) of the study.

5. Abstract: Please include the study registration number in the Abstract.

6. Author Summary: At this stage, we ask that you include a short, non-technical Author Summary of your research to make findings accessible to a wide audience that includes both scientists and non-scientists. The Author Summary should immediately follow the Abstract in your revised manuscript. This text is subject to editorial change and should be distinct from the scientific abstract. Please see our author guidelines for more information: https://journals.plos.org/plosmedicine/s/revising-your-manuscript#loc-author-summary

7. Introduction: Please expand on the rationale for the study: address past research and explain the need for and potential importance of the study.

8. Methods: Search Strategy: Please update the search to the present time (the end date is noted as October 2019).

9. Methods: Please describe how you evaluated risk of bias, including publication bias.

10. Results: Line 191: If 5 of the studies did not report a participation rate, please clarify how was it determined whether the participation rate was 50% or greater (as outlined in the exclusion criteria)?

11. Results: Line 256: Please clarify if “preferences” should be “prevalence” here.

12. References: Please check formatting of references. Please us the "Vancouver" style for reference formatting, and see our website for other reference guidelines https://journals.plos.org/plosmedicine/s/submission-guidelines#loc-references

In Ref 24, Jama should be JAMA, for example.

13. Figure 1: It may be helpful to indicate at least broadly indicate reasons for exclusion in the flow diagram at screening points.

14. Figures 2-8. For each figure, please provide a descriptive legend describing what is shown in the figure (for example, including the data points and bars, dotted line).

15. PRISMA Checklist: Thank you for including the checklist as a supporting information file. Please revise the checklist, using section and paragraph numbers to refer to locations within the text, as opposed to page numbers.

16. S3 Table: Please define all abbreviations in a legend for the Table.

17. S4 and S5 Tables: It would be helpful to include more descriptive title/legends for these tables.

18. S1 Text: Can you please explain why the results for affective disorders are being reported separately from the main text?

Comments from the reviewers:

Reviewer #1: See attachment

Michael Dewey

Reviewer #2: This manuscripts introduced a well conducted meta analysis and literature review study that aims to investigate the prevalence of mental disorders among homeless population in selected countries. The study is well conducted in that it used creditable guidelines and went through a solid process to retrieve fitting articles. It then used well recognized statistical method in data analysis. The writing of the paper in each section is clear. Discussion has its focus. The whole manuscript in writing and presenting key components of the study is cohesive and well focused. I do not see any particular errors, or mis-representations in writing. So, as for the study and its writing, I think it is publishable. So I examined this manuscript from scientific methods side of views. Below are my comments to authors and to editors. You may make revisions if you accept my comments or leave them alone since the comments are more from my personal views. 

1. While it is important to continuously build up a profile of the prevalence in literature, reporting prevalence alone or as a main content of introduction for this paper, as presented mostly in statistics terms, seems a bit thin. The whole findings of the study as presented are prevalence rates, however detailed and broken down by the type of mental disorders. 

2. Only three countries are included. I understand this is because of the articles selected through a screening process. So it is not authors' purpose to include them only. Is there a way to explain/estimate why other, economically advanced, countries are not included? 

3. Readers like me would want to see prevalence rates of each of the countries for a comparison. This will add some contextual information to the paper. 

4. As a research who has been working with homeless population, I also like to see a bit comparisons about demographic characteristics across the types of mental disorders, and countries. 

Thank you for the opportunity to review this article!

Reviewer #3: This is - in principle - a very useful and important update of the earlier review on mental disorders in people living homeless. However, the paper has two major methodological flaws related to the meta-analysis and meta-regressions that do not allow to proceed with the publication. Firstly, the variable selection procedure via p-value is as outdated as it is statistically wrong. If the authors want to stick to the model selection (there are many pros and cons related to the selection issue), they should use proper software solutions that are available in R metafor. Secondly, and also related to software issues, the authors should use a procedure that accounts for the problem of using multiple effects from the same study/publication. Again, R metafor is able to deal with this problem. In such situations, multilevel models (effects within studies/publications) should be conducted. 

In the light of these problems, I suggest a rejection and re-submission after dealing with the issues.

Reviewer #4: This is a well-structured meta-analytic review with robust methodology and profound significance for mental health research and policymaking in homeless populations. The presentation of the findings and articulation of the evidence in the context of homelessness appeared to be excellent. However, there are a few concerns that should be addressed to improve the scholarly value accuracy and implications of this manuscript.

First, the abstract should clearly mention the objective of the review before stepping into the methods. Also, the abstract should inform that it is an updated review; therefore, mentioning the previous review and specifying the timeline of the current would substantially improve the abstract.

Second, the introduction/background does not provide a clear understanding of why a review on high-income countries AND homeless people is required in the first place. It is essential to describe the "need" for a review. The authors may expand the biopsychosocial challenges associated with mental health in homeless people and how they are recognized amongst one of the most marginalized population groups in developed countries. Such explanations, among many others, are likely to inform the readers why this review is important and why we must focus on the evidence available on this critical issue. Also, multiple previous reviews on homelessness and mental health [1-4], not only the one that was published in 2007, should be discussed in the background to offer a broader understanding of the evidence landscape.

Third, the authors should specify which mental disorders were considered before screening the articles. A complete list of disorders should accompany the manuscript, perhaps as supplementary material. Also, reference#22 was cited in the methods section, which was originally published in 1985. The authors may explain why they did not use the most contemporary definitions and categories of mental disorders as stated in DSM-5 or ICD-10. An informed author may wish to understand what disorders/conditions were within and beyond the scope of the current review.

Fourth, the authors should consider sensitivity analyses to examine the effects of individual studies (and their effect sizes) on the pooled estimates for each meta-analytic model. Also, subgroup analyses in this review appeared to be limited to the risk of bias, whereas multiple subgroups could be constructed using variables such as gender (male vs. female), age (young adult vs. older adult), types of homeless (street-living vs. those in shelter homes), comorbidity (healthy vs. those with known medical conditions), geographic locations (homeless in North America vs. in Europe) etc. A set of subgroup analyses would be extremely helpful to interpret the differences in estimates in different subgroups and adopt specific measures to improve their mental health outcomes. This is important because homeless people are not homogenous [5-7]; therefore, their psychosocial stressors are likely to vary, and this must be 

recognized while estimating the burden of their mental health problems.

Fifth, the authors should provide explanations/reasons for the excluded studies, at least for those that were excluded during the full-text evaluation. That should inform how many articles were excluded for each specific reason(s).

Sixth, the discussion section of this manuscript could offer more insights on what the current evidence informs about future research priorities. One of the goals of systematic reviews and/or meta-analysis is to identify research gaps and offer guidance on critical areas that require further assessment. Moreover, the interpretation of the evidence should accompany how the practitioners and policymakers can use the evidence in practice. Prevalence estimates can inform psychosocial vulnerabilities in the population of interest and inform the need for action that may address the existing disease burden. Such avenues must be explored in the discussion of the manuscript.

Lastly, the limitations of the review must be revisited and strengthened considering the following issues: a) a limited number of databases searched (with marked overlaps between similar databases such as Medline and PubMed), and b) a lack of reporting the publication bias (can be ignored in the authors provide the funnel plots/Egger's test estimates for publication bias).

References:

1. Ayano G, Belete A, Duko B, Tsegay L, Dachew BA. Systematic review and meta-analysis of the prevalence of depressive symptoms, dysthymia and major depressive disorders among homeless people. BMJ open. 2021 Feb 1;11(2):e040061.

2. Schreiter S, Bermpohl F, Krausz M, Leucht S, Rössler W, Schouler-Ocak M, Gutwinski S. The prevalence of mental illness in homeless people in Germany: A systematic review and meta-analysis. Deutsches Aerzteblatt International. 2017 Oct;114(40):665.

3. Bassuk EL, Richard MK, Tsertsvadze A. The prevalence of mental illness in homeless children: A systematic review and meta-analysis. Journal of the American Academy of Child & Adolescent Psychiatry. 2015 Feb 1;54(2):86-96.

4. Hossain MM, Sultana A, Tasnim S, Fan Q, Ma P, McKyer EL, Purohit N. Prevalence of mental disorders among people who are homeless: An umbrella review. International Journal of Social Psychiatry. 2020 Sep;66(6):528-41.

5. Chamberlain C, MacKenzie D. Understanding contemporary homelessness: Issues of definition and meaning. Australian Journal of Social Issues. 1992 Nov;27(4):274-97.

6. Smith EM, North CS. Not all homeless women are alike: effects of motherhood and the presence of children. Community Mental Health Journal [Internet]. 1994 Dec [cited 2021 Mar 11];30(6):601-10.

7. Institute of Medicine (US) Committee on Health Care for Homeless People. Homelessness, Health, and Human Needs. Washington (DC): National Academies Press (US); 1988. 1, Who Are the Homeless? Available from: https://www.ncbi.nlm.nih.gov/books/NBK218239/

[LINK]

---

## [Decision Letter · Decision Letter 2]

22 Jul 2021

Dear Dr. Fazel,

Thank you very much for re-submitting your manuscript "The prevalence of mental disorders among homeless people in high-income countries: updated systematic review and meta-regression analysis" (PMEDICINE-D-21-00563R2) for review by PLOS Medicine.

I have discussed the paper with my colleagues and the academic editor and it was also seen again by one of the reviewers. I am pleased to say that provided the remaining editorial and production issues are dealt with we are planning to accept the paper for publication in the journal.

[LINK]

We look forward to receiving the revised manuscript by Jul 29 2021 11:59PM.   

Sincerely,

Caitlin Moyer, Ph.D.

Associate Editor 

PLOS Medicine

plosmedicine.org

Requests from Editors:

1. Title: Please capitalize the first word of the subtitle, we suggest: “The prevalence of mental disorders among homeless people in high-income countries: An updated systematic review and meta-regression analysis”

2. Abstract: Line 41-42: Please provide slightly more detail on how study quality was assessed.

3. Abstract: Line 54-56: We suggest revising to “Our findings suggest the burden of psychiatric morbidity in homeless persons is substantial…”

4. Author summary: “Why was this study done” Line 68, and Discussion: Line 436 and 512, 514, 518: We suggest replacing “the homeless” with “homeless individuals” or similar.

5. Discussion: Line 477: Please spell out “RCT” in the text, if this is the first mention of the abbreviation.

6. Page 23: Please remove the “Funding” section from the main text and ensure all information is accurately entered in the “Financial Disclosure” section of the manuscript submission form.

7. References: Please double check that all references use the "Vancouver" style for reference formatting, and see our website for other reference guidelines https://journals.plos.org/plosmedicine/s/submission-guidelines#loc-references

Ref 22, 61, 66, 84, 96: Please double check if the reference information is complete. Please check the journal title abbreviation for Reference 102.

8. Figure 2 - 8: If possible, please increase the font size if possible, particularly along the X axis labels.

9. Supporting information files: Please include a “clean” version of each table and S1 Text.

Comments from Reviewers:

Reviewer #1: The authors have addressed my points successfully.

Michael Dewey

[LINK]

---

## [Editor Report · Decision Letter 3]

2 Aug 2021

Dear Dr Fazel, 

On behalf of my colleagues and the Academic Editor, Vikram Patel, I am pleased to inform you that we have agreed to publish your manuscript "The prevalence of mental disorders among homeless people in high-income countries: An updated systematic review and meta-regression analysis" (PMEDICINE-D-21-00563R3) in PLOS Medicine.

In addition, please complete the following editorial requests:

-Abstract: Line 62: Please remove the funding information from the Abstract at line 62 (“SF is funded by the Wellcome Trust (202836/Z/16/Z)”) , as the study funding information will be included automatically from the Financial Disclosure information entered with the manuscript submission data.

-Methods: Line 213, and Supporting Information Tables: The terms gender and sex are not interchangeable; please use the appropriate term consistently throughout. The term “sex” is used in Table 1. Please replace “gender” with “sex” in this sentence in the Methods and in S4, S7, S8, S9, and S10 Tables for reporting ratio of female to all participants.

-Discussion: Line 415: Please check that the number of studies should be reported here as 39, and the number of included participants should be 8049, as indicated in the Results and Abstract. There is also a typo in Figure 1 where 38 should be 39.

-References: Please change Nov to Apr for Reference 95.

PRESS

Sincerely, 

Caitlin Moyer, Ph.D. 

Associate Editor 

PLOS Medicine